# Ocular Decompression Retinopathy after Anterior Chamber Paracentesis for Neovascular Glaucoma

**DOI:** 10.3390/medicina57101038

**Published:** 2021-09-29

**Authors:** Yung-En Tsai, Cherng-Ru Hsu

**Affiliations:** 1Department of Ophthalmology, Kaohsiung Armed Forces General Hospital, Kaohsiung 802, Taiwan; zun7511@gmail.com; 2Department of Ophthalmology, Tri-Service General Hospital, National Defense Medical Center, Taipei 114, Taiwan

**Keywords:** ocular decompression retinopathy, anterior chamber paracentesis, neovascular glaucoma

## Abstract

Ocular decompression retinopathy (ODR) is a rare complication associated with intraocular pressure (IOP)-lowering interventions. We report a case of neovascular glaucoma in the left eye with marked IOP elevation (33 mmHg in the left eye). The IOP in the left eye did not improve despite medical treatment. Paracentesis of the left eye was then performed, and the IOP in the left eye decreased to 9 mmHg. One day after the procedure, several intraretinal hemorrhages, Roth spots, and subhyaloid hemorrhages appeared in the fundus of the left eye. Left eye retinopathy was likely ODR because of the rapid decline in IOP.

## 1. Introduction

Ocular decompression retinopathy (ODR) is a rare complication, first described by Fechtner et al. in 1992, in association with intraocular pressure (IOP)-lowering interventions [1]. ODR occurred after medical or surgical procedures for various types of glaucoma. Approximately half of ODR cases occurred after trabeculectomy [2]. Other surgical procedures included glaucoma drainage implant insertions, trabeculotomy, iridotomy, iridoplasty, anterior chamber paracentesis, orbital decompression, phacoemulsification and vitrectomy. It is characterized by multiple retinal hemorrhages and optic nerve changes. Hemorrhages occur in all retinal layers, and most were intra-retinal and subhyaloid [2]. We report the case of a 66-year-old man who developed ODR after anterior chamber paracentesis for neovascular glaucoma (NVG).

## 2. Case Report

A 66-year-old man with type 2 diabetes mellitus presented with a 3-day history of painful blurry vision in his left eye. The left eye examination showed microcystic edema of the cornea, hyphema with iris neovascularization, and marked IOP elevation (11 mmHg in the right eye and 33 mmHg in the left eye, measured by pneumatic tonometer) (Figure 1). He was diagnosed with NVG in the left eye. The visual acuity of the left eye was 20/200 at the time of presentation. The anterior segment examination of the right eye was unremarkable. Fundus examination of the left eye was obscured by microhyphema and microcystic corneal edema, and the right eye was diagnosed with non-proliferative diabetic retinopathy. The IOP of the left eye did not improve despite maximal intravenous mannitol usage, the combination of topical brinzolamide/timolol in the left eye twice a day and 500 mg of acetazolamide orally twice a day. Paracentesis of the left eye was then performed, and the IOP in the left eye decreased to 9 mmHg. One day after the procedure, the cornea became clear. Dilated fundus examination revealed several intraretinal hemorrhages and subhyaloid hemorrhages in the posterior pole and mid-peripheral retina (Figure 2) and Roth spots in the peripheral retina of the left eye. Optical coherence tomography of the left eye revealed a superficial retinal hemorrhage and dense pre-macular hemorrhage (Figure 3). Fluorescein angiography of the left eye showed multiple areas of fluorescein blockage, corresponding to hemorrhage in the fundus. Scattered hyperfluorescent microaneurysms were also observed in the fundus of both eyes (Figure 4). The visual acuity of the left eye decreased to hand motion. The peripheral retina of left eye showed multiple non-perfusion areas in fluorescein angiography (Figure 5), indicated retinal ischemia. Panretinal photocoagulation was performed. One week later, an intravitreal anti-vascular endothelial growth factor agent was injected to treat NVG. Topical timolol was prescribed to control the IOP. During the follow-up fundus examination three months later, retinal hemorrhages were minimally reabsorbed. His best-corrected visual acuity of the left eye improved to 20/400. The left eye retinopathy was likely ODR because of the rapid decline in IOP.

## 3. Discussion

ODR is a rare postoperative complication following a rapid decrease in IOP. ODR was initially described as multiple retinal hemorrhages following glaucoma filtering procedures, but its association between other surgical and medical procedures have been reported in previous research. The diagnosis of ODR is mainly based on clinical findings. Approximately 80% of ODR patients are asymptomatic [2]. Its symptoms include decreased vision, central scotoma, and floaters [2]. The fundus findings in ODR include hemorrhages in all retinal layers and optic nerve changes. There is no consensus regarding the pathophysiology of ODR. Mechanisms involving mechanical and vascular etiologies have been proposed. The anatomical mechanism is based on a forward displacement of the lamina cribrosa, causing blockage or decreased axonal transport, central retinal vein compression, and retinal hemorrhages, mimicking retinal vein thrombosis. The vascular mechanism involves an abrupt increase in retinal intravascular flow due to hypotony and defective blood vessel autoregulation secondary to long-standing glaucoma. This ultimately results in retinal hemorrhage [1,3].

Our patient had type 2 diabetes mellitus with poor glycemic control and had elevated IOP of the left eye for at least 3 days. There may have been a loss of autoregulation of retinal capillaries. As seen in poorly controlled diabetes mellitus, the loss of retinal capillary autoregulation increases the risk for retinal capillary leakage when there is a sudden increase in retinal blood flow. With the sudden decrease in IOP after anterior chamber paracentesis, an increase in arterial blood flow may have overwhelmed the capillary resistance, leading to retinal hemorrhages. To avoid ODR, gradual decrease in preoperative IOP with the appropriate anti-glaucomatous medication use preoperatively and intraoperative IOP with slow and deliberate procedure [4]. In most of the cases, ODR follows a benign course and visual acuity usually returned to preoperative levels without treatment. Although it is a self-limiting condition, a few patients may require vitrectomy for nonresolving vitreous hemorrhage [2].

There are some cases of ODR that occurred after anterior chamber paracentesis in primary open-angle glaucoma [5] and uveitic glaucoma [6,7]. To the best of our knowledge, only one case of ODR occurring after anterior chamber paracentesis in NVG has been reported in the English literature [8]. In our patient, the eye presented with NVG and persistent elevated IOP. After anterior chamber paracentesis, a sudden drop in IOP was noted and ODR occurred after 1 day. Our case demonstrates the rapid onset of ODR after anterior chamber paracentesis in NVG following IOP reduction.

In conclusion, ODR can occur after anterior chamber paracentesis in cases of NVG where there is a rapid decline in IOP and a gradual decrease in intraoperative IOP is recommended to minimize the risk of ODR [4].

## Figures and Tables

**Figure 1 medicina-57-01038-f001:**
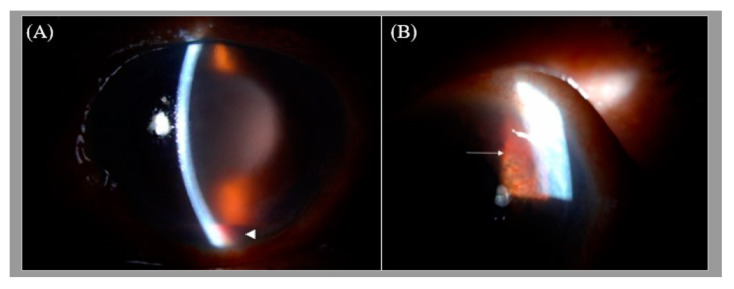
(**A**) The slit lamp image of the left eye showing microcystic edema in the cornea and hyphema. (**B**) Iris neovascularization (white arrow). The hazy view is secondary to microcystic corneal edema associated with increased intraocular pressure in this case of neovascular glaucoma.

**Figure 2 medicina-57-01038-f002:**
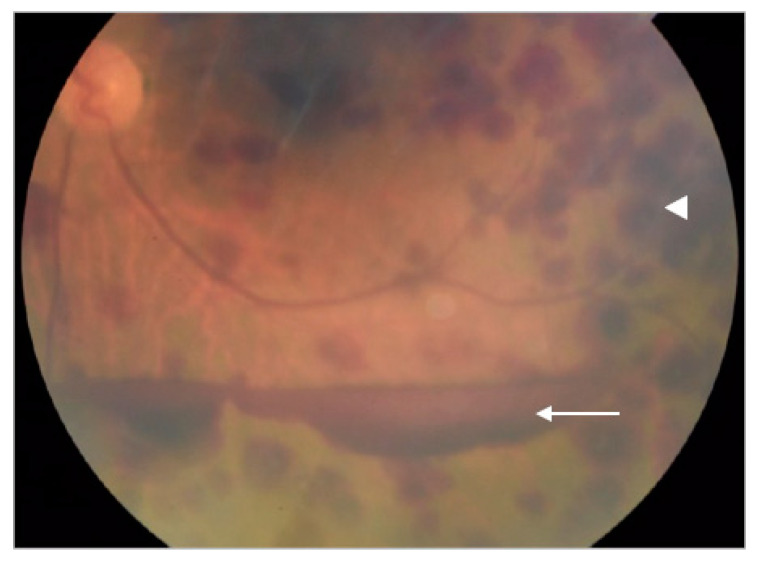
Fundus photography of the left eye showing several intraretinal hemorrhages (white arrowhead) and subhyaloid hemorrhages (white arrow) in the posterior pole and mid-peripheral retina.

**Figure 3 medicina-57-01038-f003:**
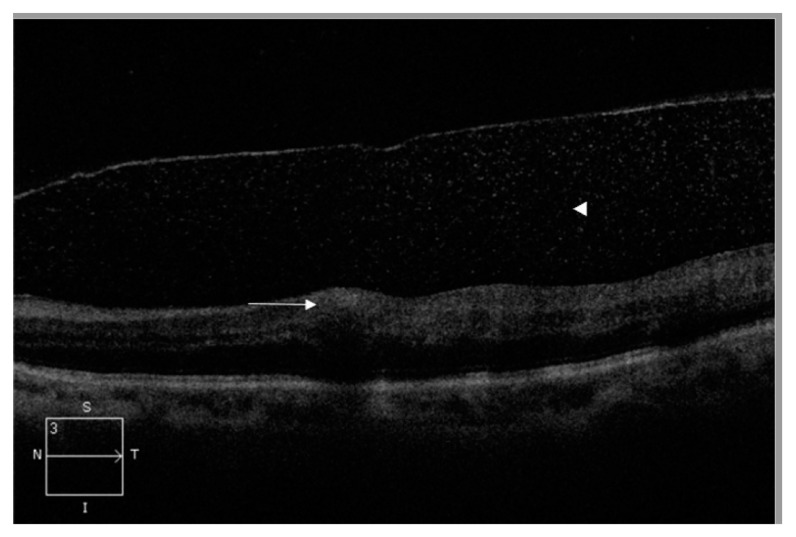
Optical coherence tomography image of the left eye demonstrating superficial retinal hemorrhage (white arrow) and dense pre-macular hemorrhage (white arrowhead).

**Figure 4 medicina-57-01038-f004:**
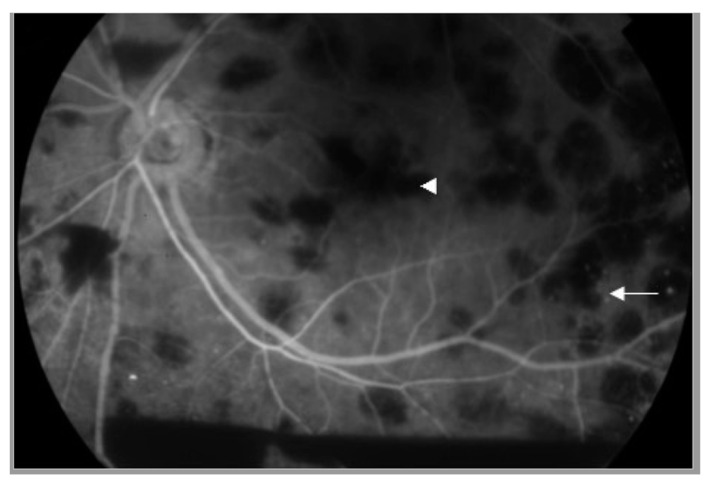
Fluorescein angiography image of the left eye showing multiple areas of blockage of fluorescein (white arrowhead). Scattered hyperfluorescent microaneurysms (white arrow) are also noted.

**Figure 5 medicina-57-01038-f005:**
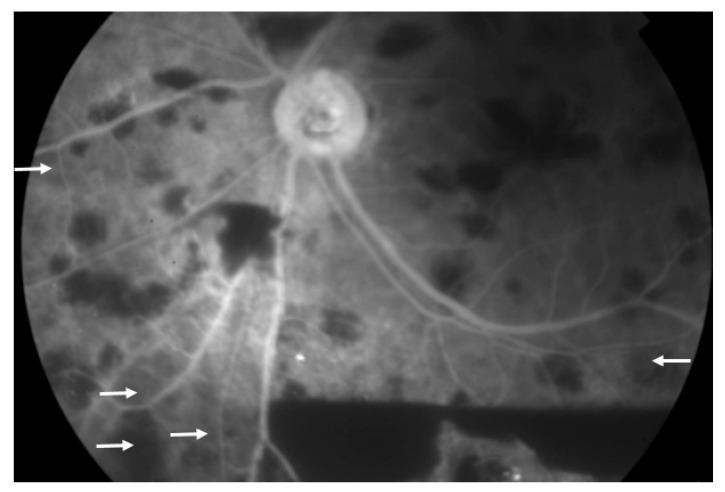
Fluorescein angiography image of the left eye of inferior periphery showing multiple non-perfusion areas (white arrows), indicated retinal ischemia.

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
