# Peer review of "Ocular Decompression Retinopathy after Anterior Chamber Paracentesis for Neovascular Glaucoma"

_medicina, 2021, doi:10.3390/medicina57101038_

Round 1

Reviewer 1 Report

It is a well-written and straightforward case report. However, control (or pre ODR) images are missing and multiple other case reports on ODR after AC paracentesis exist that can be cited.

Examples are:

a- https://www.ncbi.nlm.nih.gov/pmc/articles/PMC6760348/

b- https://pubmed.ncbi.nlm.nih.gov/18728618/

c- https://pubmed.ncbi.nlm.nih.gov/16892651/

Also, it would be great if some thoughts/ideas on ODR prevention/treatment are included in the discussion. 

Some other points:

Line 41: Providing information about the failed medications would be informative.

Line 81-82: Provide references for these cases of ODR after NVGLine 83-85: Seems like redundant information

Figure 1: Mention the technique used to obtain the images in the legend

Figure 2: Use arrows (or other symbols) to point out hemorrhages, Roth spots, and subhyaloid hemorrhages.

Author Response

Thank you for allowing us to submit a revised draft of our manuscript entitled,” Ocular decompression retinopathy after anterior chamber paracentesis for neovascular glaucoma.” to MEDICINA. The previous manuscript ID is: medicina-1371745. We also appreciate the time and effort you and each of the Reviewers have dedicated to providing constructive feedback.

Our responses to the Reviewers’ comments and suggestions are described below in a point-to-point manner. Appropriate changes, suggested by the Reviewers, has been introduced to the manuscript.

Point 1: Control (or pre ODR) images are missing.

Response 1: Because this patient visited our emergency room presented with persisted IOP elevation and cornea edema, we decided to alleviate his symptoms first. Pharmacologic pupil dilation may cause IOP elevation, so we didn’t perform dilated fundus examination initially. Thus, we didn’t have his pre ODR images before the onset of the disease.

Point 2: Multiple other case reports on ODR after AC paracentesis exist that can be cited.

Response 2: We cited three case reports on ODR after AC paracentesis in primary open-angle glaucoma and uveitic glaucoma in discussion.

Page 2 Line 86, 87: “There are some cases of ODR occurred after anterior chamber paracentesis in primary open-angle glaucoma[5] and uveitic glaucoma[6,7].”

References

  1. Jamie P, David F. Immediate manifestation of ocular decompression retinopathy following anterior chamber paracentesis. Case Rep Ophthalmol 2019;10:287–291.
  2. Sang JL, Jung JL, Shin DK. Multiple retinal hemorrhage following anterior chamber paracentesis in uveitic glaucoma. KJO 2006;20(2):128-130.
  3. Sunil KR, Paul BG, Ross BL, et al. Ocular decompression retinopathy after anterior chamber paracentesis for uveitic glaucoma. Retina.2009 Feb;29(2):280-1.

Point 3: It would be great if some thoughts/ideas on ODR prevention/treatment are included in the discussion.

Response 3: We added some thoughts and ideas on prevention and treatment of ODR in Page 2 line 80-85.

Page 2 line 80-85: “To avoid ODR, gradual decrease of preoperative IOP with the appropriate anti-glaucomatous medication use preoperatively and intraoperative IOP with slow and deliberate procedure[4]. In most of the cases of ODR follows a benign course and visual acuity usually returning to preoperative levels without treatment. Although it is a self-limiting condition, a few patients may require vitrectomy for nonresolving vitreous hemorrhage[2].”

Point 4: Line 41: Providing information about the failed medications would be informative.

Response 4: We provided the failed medications with “maximal intravenous mannitol usage, the combination of topical brinzolamide/timolol in the left eye twice a day and 500 mg of acetazolamide orally twice a day” in Page 1 line 42, 43&44.

Point 5: Line 81-82: Provide references for these cases of ODR after NVG.

Response 5: We provided a reference [8] for the case of ODR in NVG.

References

  1. Semra K, Defne K, Zeynep D, et al. Ocular decompression retinopathy following anterior chamber paracentesis in a patient with neovascular glaucoma associated with diabetic retinopathy. Research 2014;1:975.

Point 6: Line 83-85: Seems like redundant information.

Response 6: We shortened and simplified this sentence in Page 2 line 89, 90&91.

Page 2 line 89, 90&91: “In our patient, the eye presented with NVG and persistent elevated IOP. After anterior chamber paracentesis, a sudden dropped of IOP was noted and ODR occurred after 1 day.”

Point 7: Figure 1: Mention the technique used to obtain the images in the legend.

Response 7: We used slit lamp biomicroscopy with integration of a digital camera technique to obtain the images and added in the figure legend of Figure 1.

Figure legend of Figure 1: ” (A) The slit lamp image of the left eye showing microcystic edema in the cornea and hyphema. (B) Iris neovascularization (white arrow). The hazy view is secondary to microcystic corneal edema associated with increased intraocular pressure in this case of neovascular glaucoma.”

Point 8: Figure 2: Use arrows (or other symbols) to point out hemorrhages, Roth spots, and subhyaloid hemorrhages.

Response 8: In the Figure 2, which showed several intraretinal hemorrhages and subhyaloid hemorrhages in the posterior pole and mid-peripheral retina and we added arrowhead and arrow to point out, respectively. However, Roth spots showed in the peripheral retina of the left eye. Thus, we deleted the “Roth spots” mentioned in the figure legend of Figure 2 and added at Page 2 line 48.

Figure legend of Figure 2: ” Fundus photography of the left eye showing several intraretinal hemorrhages (white arrowhead) and subhyaloid hemorrhages (white arrow) in the posterior pole and mid-peripheral retina.”

Page 2 line 46-48: “Dilated fundus examination revealed several intraretinal hemorrhages and subhyaloid hemorrhages in the posterior pole and mid-peripheral retina (Figure 2) and Roth spots in the peripheral retina of the left eye.”

Thanks a lot again and your comments and suggestions really helped us to improve our manuscript better. We hope that our manuscript will be acceptable for publication in MEDICINA.

We look forward to hearing from you.

Sincerely,

Cherng-Ru Hsu

Department of Ophthalmology, Tri-Service General Hospital, National Defense Medical Center, Taipei, Taiwan.

No. 325, Chenggong Rd., Sec. 2, Neihu, Taipei 114, Taiwan.

Tel: +886-972313173

Fax: +886-774-052-31

E-mail: josephinesheu@gmail.com

Reviewer 2 Report

  • How did you measure the intraocular pressure in this edematous cornea? . You didn’t mention the method.
  • Was  33 mmhg intraocular pressure responsible for this edematous cornea? . I think corneal edema occurs with IOP more than 33 mmhg.
  • Page 1 line 39, 40. “Fundus examination of the left eye was obscured by hyphema” but in figure 1 hyphema was in lower third below pupil.
  • Page 1 line 42, 43. “Paracentesis of the left eye was then performed, and the IOP in the left eye decreased to 9 mmHg. One day after the procedure, the cornea became clear”.

The IOP was not controlled medically and with paracentesis decreased to 9 mmhg and remains low for the 2nd day?. Usually in similar cases IOP start to increase again few hours after the paracentesis.

  • Page 2 line 50. “The visual acuity of the left eye decreased to hand motion”. Why? You mentioned that the cornea became clear after paracentesis and in figure 2 subhyaloid hemorrhage is in lower part and the macula is affected only by retinal hemorrhages not responsinle for hand motion.

  • Page 2 line 50,51&52. “Complete panretinal photocoagulation was performed”. How was it performed in the extensive retinal hemorrhages which appear in figure 2 and figure 4?

  • Page 2 line 50,51&52. “Complete panretinal photocoagulation was performed. One week later, an intravitreal anti-vascular endothelial 51 growth factor agent was injected to treat NVG”. Why? There was no retinal ischemia or signs of proliferative diabetic retinopathy in figure 2 and 4.
  • Any blood diseases or history of trauma in this patient?

What was the cause of neovascular glaucoma in this patient?

Author Response

Thank you for allowing us to submit a revised draft of our manuscript entitled,” Ocular decompression retinopathy after anterior chamber paracentesis for neovascular glaucoma.” to MEDICINA. The previous manuscript ID is: medicina-1371745. We also appreciate the time and effort you and each of the Reviewers have dedicated to providing constructive feedback.

Our responses to the Reviewers’ comments and suggestions are described below in a point-to-point manner. Appropriate changes, suggested by the Reviewers, has been introduced to the manuscript.

Point 1: How did you measure the intraocular pressure in this edematous cornea? You didn’t mention the method.

Response 1: We used pneumatic tonometer to measure the intraocular pressure. And we added to our manuscript in Page 1 line 37.

Point 2: Was 33 mmHg intraocular pressure responsible for this edematous cornea? I think corneal edema occurs with IOP more than 33 mmHg.

Response 2: Intraocular pressure would be affected by many factors, such as corneal edema, corneal thickness, daily variation…etc. We didn’t measure this patient’s corneal thickness and it could be possible that we had underestimated his intraocular pressure. 

Point 3: Page 1 line 39, 40. “Fundus examination of the left eye was obscured by hyphema” but in figure 1 hyphema was in lower third below pupil.

Response 3: Although hyphema was in lower third below pupil, prominent microhyphema was found in the anterior chamber. In addition, microcystic cornea edema further hinder the detail examination of the fundus. We replaced “hyphema” with “microhyphema and microcystic corneal edema” in Page 1 line 40, 41.

Point 4: Page 1 line 42, 43. “Paracentesis of the left eye was then performed, and the IOP in the left eye decreased to 9 mmHg. One day after the procedure, the cornea became clear.” The IOP was not controlled medically and with paracentesis decreased to 9 mmHg and remains low for the 2nd day?. Usually in similar cases IOP start to increase again few hours after the paracentesis.

Response 4: Before paracentesis of the left eye, we prescribed maximal intravenous mannitol usage, the combination of topical brinzolamide/timolol in the left eye twice a day and 500 mg of acetazolamide orally twice a day. After paracentesis of the left eye, we kept the combination of topical brinzolamide/timolol in the left eye twice a day and 500 mg of acetazolamide orally twice a day. We added this information in Page 1 line 42, 43&44.

Point 5: Page 2 line 50. “The visual acuity of the left eye decreased to hand motion”. Why? You mentioned that the cornea became clear after paracentesis and in figure 2 subhyaloid hemorrhage is in lower part and the macula is affected only by retinal hemorrhages not responsible for hand motion.

Response 5: Except retinal hemorrhages in macula, the fluorescein angiography image of the left eye also showing enlargement of foveal avascular zone. However, the quality of fluorescein angiography image of the left eye was poor due to microhyphema. Besides, focal ellipsoid zone disruption and inner retinal layer disorganization were found in the macular OCT scan (Figure 3). These may explain why the visual acuity of the left eye decreased to hand motion.

Point 6: Page 2 line 50, 51&52. “Complete panretinal photocoagulation was performed”. How was it performed in the extensive retinal hemorrhages which appear in figure 2 and figure 4?

Response 6: We tried our best to perform panretinal photocoagulation. When encountered the area of retina blocked by hemorrhages, we used long-wavelength yellow or red laser. We replaced “Complete panretinal photocoagulation” with “Panretinal photocoagulation” in Page 2 line 53 because some laser spots may fail to perform on the area blocked by retinal hemorrhages.

Point 7: Page 2 line 50, 51&52. “Complete panretinal photocoagulation was performed. One week later, an intravitreal anti-vascular endothelial 51 growth factor agent was injected to treat NVG”. Why? There was no retinal ischemia or signs of proliferative diabetic retinopathy in figure 2 and 4.

Response 7: This patient had underlying disease with type 2 diabetes mellitus with poor control and the fundus examination of his right eye revealed non-proliferative diabetic retinopathy. His left eye may have diabetic retinopathy, though the fundus examination obscured by microhyphema and microcystic corneal edema initially. Besides, the left eye examination showed iris neovascularization, the vascular endothelial growth factor may exist in the posterior segment of his left eye. Therefore, we arranged injection of intravitreal anti-vascular endothelial growth factor agent for this patient.

Point 8: Any blood diseases or history of trauma in this patient?

Response 8: Initially, in emergency room, we checked complete blood count with differential count and other laboratory survey, such as blood coagulation test, renal function, liver function and electrolyte, all were in normal range, except high blood glucose was found. He also didn’t have any trauma history.

Point 9: What was the cause of neovascular glaucoma in this patient?

Response 9: This patient had underlying disease with type 2 diabetes mellitus with poor control. We highly suspected the cause of neovascular glaucoma in this patient was type 2 diabetes mellitus.

Thanks a lot again and your comments and suggestions really helped us to improve our manuscript better. We hope that our manuscript will be acceptable for publication in MEDICINA.

We look forward to hearing from you.

Sincerely,

Cherng-Ru Hsu

Department of Ophthalmology, Tri-Service General Hospital, National Defense Medical Center, Taipei, Taiwan.

No. 325, Chenggong Rd., Sec. 2, Neihu, Taipei 114, Taiwan.

Tel: +886-972313173

Fax: +886-774-052-31

E-mail: josephinesheu@gmail.com

Round 2

Reviewer 2 Report

Dear authors in revised manuscript you didnt add figures showing peripheral retinal ischaemia. To support the management you performed.

  • Page 2 line 50,51&52. “Complete panretinal photocoagulation was performed. One week later, an intravitreal anti-vascular endothelial 51 growth factor agent was injected to treat NVG”. Why? There was no retinal ischemia or signs of proliferative diabetic retinopathy in figure 2 and 4.
  • What was the cause of neovascular glaucoma in this patient?

Author Response

Dear Reviewer:

Thank you for allowing us to submit a revised draft of our manuscript entitled,” Ocular decompression retinopathy after anterior chamber paracentesis for neovascular glaucoma.” to MEDICINA. The previous manuscript ID is: medicina-1371745. We also appreciate the time and effort you have dedicated to providing constructive feedback.

Our responses to the Reviewers’ comments and suggestions are described below in a point-to-point manner. Appropriate changes, suggested by the Reviewers, has been introduced to the manuscript.

Point 1: You didn’t add figures showing peripheral retinal ischaemia. To support the management you performed.

Response 1: We added Figure 5 of fluorescein angiography image of the left eye of inferior periphery showing multiple non-perfusion areas (white arrows), indicated retinal ischemia. We also added this information in Page 2 line 53, 54&55 with “The peripheral retina of left eye showed multiple non-perfusion areas in fluorescein angiography (Figure 5), indicated retinal ischemia.”

Point 2: Page 2 line 50, 51&52. “Complete panretinal photocoagulation was performed. One week later, an intravitreal anti-vascular endothelial 51 growth factor agent was injected to treat NVG”. Why? There was no retinal ischemia or signs of proliferative diabetic retinopathy in figure 2 and 4.

Response 2: In Figure 5, there were multiple non-perfusion areas, which indicated the retinal ischemia that corresponding to the underlying diabetic retinopathy. Because of diabetic retinopathy with retinal ischemia, the vascular endothelial growth factor may exist in the posterior segment of his left eye. Besides, the left eye examination showed iris neovascularization. Therefore, we arranged injection of intravitreal anti-vascular endothelial growth factor agent for this patient.

Point 3: What was the cause of neovascular glaucoma in this patient?

Response 3: There were multiple non-perfusion areas shown in fluorescein angiography, which indicated the retinal ischemia that corresponding to the underlying diabetic retinopathy. Although we did not perform the gonioscopy to identify the neovascularization in anterior chamber angle, the neovascularization on the iris surface was essential for initial diagnosis of neovascular glaucoma.

Thanks a lot again and your comments and suggestions really helped us to improve our manuscript better. We hope that our manuscript will be acceptable for publication in MEDICINA.

We look forward to hearing from you.

Sincerely,

Cherng-Ru Hsu

Department of Ophthalmology, Tri-Service General Hospital, National Defense Medical Center, Taipei, Taiwan.

No. 325, Chenggong Rd., Sec. 2, Neihu, Taipei 114, Taiwan.

Tel: +886-972313173

Fax: +886-774-052-31

E-mail: josephinesheu@gmail.com